# Potential for Drug Repositioning of Midazolam as an Inhibitor of Inflammatory Bone Resorption

**DOI:** 10.3390/ijms25147651

**Published:** 2024-07-12

**Authors:** Hiroko Harigaya, Risako Chiba-Ohkuma, Takeo Karakida, Ryuji Yamamoto, Keiko Fujii-Abe, Hiroshi Kawahara, Yasuo Yamakoshi

**Affiliations:** 1Department of Dental Anesthesiology, School of Dental Medicine, Tsurumi University, 2-1-3 Tsurumi, Tsurumi-ku, Yokohama 230-8501, Japan; pd21008@stu.tsurumi-u.ac.jp (H.H.); fujii-keiko@tsurumi-u.ac.jp (K.F.-A.); kawahara-h@tsurumi-u.ac.jp (H.K.); 2Department of Biochemistry and Molecular Biology, School of Dental Medicine, Tsurumi University, 2-1-3 Tsurumi, Tsurumi-ku, Yokohama 230-8501, Japan; chiba-r@tsurumi-u.ac.jp (R.C.-O.); karakida-t@tsurumi-u.ac.jp (T.K.); yamamoto-rj@tsurumi-u.ac.jp (R.Y.)

**Keywords:** midazolam, drug repositioning, inflammation, bone resorption, reactive oxygen species, signal transduction

## Abstract

Drug repositioning is a method for exploring new effects of existing drugs, the safety and pharmacokinetics of which have been confirmed in humans. Here, we demonstrate the potential drug repositioning of midazolam (MDZ), which is used for intravenous sedation, as an inhibitor of inflammatory bone resorption. We cultured a mouse macrophage-like cell line with or without MDZ and evaluated its effects on the induction of differentiation of these cells into osteoclasts. For in vivo investigations, we administered lipopolysaccharide (LPS) together with MDZ (LPS+MDZ) to the parietal region of mice and evaluated the results based on the percentage of bone resorption and calvaria volume. Furthermore, we examined the effects of MDZ on the production of reactive oxygen species (ROS) in cells and on its signaling pathway. MDZ inhibited osteoclast differentiation and bone resorption activity. In animal studies, the LPS+MDZ group showed a decreasing trend associated with the rate of bone resorption. In addition, the bone matrix volume in the LPS+MDZ group was slightly higher than in the LPS only group. MDZ inhibited osteoclast differentiation by decreasing ROS production and thereby negatively regulating the p38 mitogen-activated protein kinase pathway. Thus, we propose that MDZ could potentially be used for treating inflammatory bone resorption, for example, in periodontal disease.

## 1. Introduction

Bone undergoes a continuous process of remodeling, which involves resorption, in which old bone is dissolved gradually, and formation, during which new bone replaces the resorbed bone. This remodeling is performed in a regulated system within which osteoclasts, which are responsible for bone resorption, and osteoblasts, which are responsible for bone formation, are closely associated [1]. In normal periodontal tissues, bone resorption and formation are balanced by regulating the promotion and suppression of osteoclast formation [2,3].

In the onset and progression of these bone resorption- and bone formation-related processes, various cellular and molecular elements, ranging from immune cells to growth factors and proteins, play multifactorial roles and contribute to the balance and progression of the cellular and molecular dynamics underlying bone resorption and bone formation [4].

Periodontal disease is an inflammatory disease mainly caused by oral bacteria present in plaques and is broadly classified into gingivitis and periodontitis [5]. If left untreated, gingivitis progresses to periodontitis, which causes the destruction of periodontal tissues, including the cementum, periodontal ligament, and alveolar bone. Basic periodontal therapy and periodontal surgery are the two main treatments for periodontal disease [6]. Treatment of periodontitis is a multi-step process, and the first step in nonsurgical periodontal therapy includes plaque control above the gingival margin, followed by subgingival scaling and root planing to remove biofilm, endotoxins, and calculus [7]. Periodontal surgery may be indicated for the removal of biofilms in areas such as severe periodontal pockets where removal with basic periodontal treatment is not feasible or when complex alveolar bone loss resulting from periodontal disease is observed. During this procedure, the regeneration of lost periodontal tissue is promoted by the concomitant use and application of periodontal tissue regeneration materials. Materials used for periodontal tissue regeneration therapy regenerate the lost alveolar bone and periodontal ligament by promoting the growth of periodontal tissue cells [8,9,10] or bone formation when implanted into bone defects caused by periodontal disease [11,12,13,14].

Drug repositioning is a method for exploring the new effects of existing drugs, the safety and pharmacokinetics of which have been confirmed in humans, with the aim of expanding their potential for practical applications. The certainty regarding the safety and pharmacokinetics of the drug at the clinical level and low costs owing to the reliance on existing data are among the prominent advantages of drug repositioning. Midazolam (MDZ) is a synthetic imidazobenzodiazepine derivative that exhibits pharmacological effects, such as hypnotic, sedative, anesthetic, anxiolytic, muscle relaxant, and anticonvulsant effects [15]. MDZ has also been shown to promote the differentiation of porcine dental pulp-derived cell lines into odontoblasts [16]. In combination with bone morphogenic protein-2, MDZ promotes the differentiation of immortalized mouse myoblast cell lines into osteoblasts and induces calcification [17]. These findings indicate the potential of repositioning MDZ as an osteogenesis-promoting drug. In addition, MDZ has been shown to suppress the production of interleukin-6 in human peripheral blood mononuclear cells, potentially suppressing immune and inflammatory responses [18]. However, the effect of MDZ on bone resorption remains unclear.

Here, we examined the effects of MDZ on bone resorption at the biochemical, cellular, and molecular morphological levels using cell-based in vitro and animal experiments.

## 2. Results

### 2.1. Effect of MDZ on the Differentiation of RAW264 Cells into Osteoclasts

Midazolam (MDZ) (molecular weight, 325.77 g/mol) is a short-acting benzodiazepine derivative with an imidazole structure (Figure 1A). We investigated the effects of MDZ on the differentiation of mouse macrophage cell line (RAW264 cells) into osteoclasts. The tartrate-resistant acid phosphatase (TRAP) activity in RAW264 cells cultured in the presence of receptor activator of NF-kappa B ligand (RANKL) alone, without the addition of MDZ (i.e., 0 μM MDZ), was dramatically upregulated compared with that in the control cells cultured in the absence of RANKL (RANKL(−)). MDZ reduced the TRAP activity levels in a concentration-dependent manner, particularly at 10, 20, and 40 μM, with a 1.58–12.2-fold decrease in activity relative to that at 0 μM MDZ (i.e., control) (Figure 1B). Setting the TRAP activity at 0 μM MDZ as the maximum (100%), the concentration (EC_50_) at which MDZ had half the effect on the differentiation of RAW264 cells into osteoclasts was 11.8 μM (Figure 1C). The concentration-dependent suppression of differentiation and fusion in osteoclasts were evident in TRAP staining; multinucleated osteoclasts were not observed at 20 and 40 μM MDZ (Figure 1D).

### 2.2. Effect of MDZ on the Expression of Osteoclast Differentiation Marker Genes

To corroborate the results of cell biology experiments at the molecular level, we examined the effects of MDZ on the expression of osteoclast differentiation marker genes. Compared with the RANKL(−) control, the mRNA levels of three osteoclast differentiation marker genes were dramatically upregulated (11.3-, 950-, and 1585-fold for nuclear factor of activated T cells 1 (*Nfatc1*), tartrate-resistant acid phosphatase (*Trap*), and cathepsin K (*Ctsk*), respectively) upon the addition of RANKL (i.e., in the RANKL(+) control). However, MDZ treatment decreased the mRNA levels of these genes in a concentration-dependent manner; in particular, the mRNA levels in cells treated with 20 μM MDZ were significantly lower (20.2-, 11.6-, and 35.0-fold for *Nfatc1*, *Trap*, and *Ctsk*, respectively) than those in the RANKL(+) control (Figure 2).

### 2.3. Effect of MDZ on Osteoclast-Mediated Bone Resorption

To evaluate the potential efficacy of MDZ in osteoclast-mediated bone resorption, we cultured RAW264 cells on calcium phosphate (CaP)-coated plates in the presence or absence of MDZ and measured the area of the bone resorption pit on the fourth day after the addition of MDZ. As expected, we observed black pits on CaP with osteoclasts exposed to 0 (i.e., RANKL(+) control) to 10 μM MDZ. In contrast, only a few pits were formed at 20 and 40 μM MDZ (Figure 3A). Moreover, the areas of resorption pit at 20 and 40 μM MDZ were approximately 4.13–7.75-fold lower than that at 0 μM MDZ (Figure 3B). These results indicate that MDZ can suppress the formation of multinucleated osteoclasts and resorption of bone matrix.

### 2.4. Animal Experiments

#### 2.4.1. Morphological Assessment of the Effect of MDZ on Lipopolysaccharide (LPS)-Induced Calvarial Mouse

We established a calvarial mouse model to evaluate the efficacy of MDZ in preventing pathological bone resorption. A total of 18 mice were divided into three groups (*n* = 6 in each group). During the experiment, all mice survived. The body weights of the animals during the inoculation period are shown in Figure 4A. Unlike in the control group, the body weight started to decrease on the second day of inoculation in the LPS-only and LPS+MDZ groups, and the maximum decrease was observed in the LPS-only group on the seventh day. Micro-computed tomography (micro-CT) scanning and three-dimensional (3D) reconstruction revealed obvious bone resorption in the LPS-injected calvarias, but the LPS+MDZ group showed limited calvarial resorption (Figure 4B and Appendix A). TRAP staining also showed high expression of TRAP in the lambdoidal, coronal, and sagittal sutures of LPS-injected calvarias, whereas the LPS+MDZ combination reduced the expression of TRAP in the lambdoidal sutures and part of the sagittal sutures (Figure 4C and Appendix A).

#### 2.4.2. Validation of Suture Width and Bone Matrix Volume on Bone Destruction in LPS-Induced Calvarial Mice

To quantitatively evaluate the morphological observations described above, we selected ten locations on the lambdoidal suture (Figure 5A and Appendix A) and three locations on the sagittal suture (Figure 5B and Appendix A) and measured the average width of each group. For the lambdoidal suture, the average width in the LPS-only group was 2.38-fold higher than that in the control group, whereas the average width in the LPS+MDZ group was almost the same as that in the control group (Figure 5C). For the sagittal suture, the average width in both the LPS-only and LPS+MDZ groups was significantly higher than that in the control group (3.15-fold in the LPS-only group and 2.11-fold in the LPS+MDZ group). Moreover, the average width in the LPS+MDZ group was lower than that in the LPS-only group (Figure 5D).

Next, we determined the bone matrix volume around both lambdoidal and coronal sutures (Figure 6). For each mouse, we created a block for the lambdoidal (Figure 6A–C) and coronal sutures (Figure 6D–F) and measured the bone matrix volume area for each block in each suture. Figure 6G and Figure 6H show a comparative analysis of the average volume of lambdoidal and coronal sutures obtained from six mice. In the lambdoidal suture group, the average volume in the LPS-only group was approximately 12% lower than that in the control group, whereas the average volume in the LPS+MDZ group was slightly different from that in the control group (Figure 6G). For the coronal suture, the average volume in the LPS-only group was significantly lower (approximately 72%) than that in the control group. Although the average volume in the LPS+MDZ group was approximately 43% lower than that in the control group, the rate of decrease was less than that in the LPS-only group (Figure 6H). These results were consistent with the morphological observations made in the micro-CT analysis. All of the abovementioned data indicate that the calvarial model was successfully established.

### 2.5. Elucidation of the Mechanism of MDZ-Induced Inhibition of Osteoclast Differentiation

We determined the mechanism by which MDZ inhibits osteoclast differentiation. We focused on reactive oxygen species (ROS) produced in the mitochondria because ROS regulate the p38 mitogen-activated protein kinase (p38 MAPK) pathway, which is important for osteoclast differentiation. Figure 7A shows the effect of MDZ on ROS production during the differentiation of RAW264 cells. Antimycin A, which is a fungal antibiotic that inhibits complex III of the mitochondrial electron transport system and produces ROS when applied to cells, was used as a positive control [19]. The ROS production in RAW264 cells cultured with antimycin A, RANKL alone, or LPS alone was higher (1.72-, 1.55-, and 1.80-fold for RANKL alone, LPS alone, and antimycin A, respectively) than that in the control cells. However, the ROS production in cells cultured in the presence of MDZ was suppressed to levels similar to that found in the control cells. To further investigate the effect of MDZ on p38 phosphorylation and ROS production, we performed Western blot analysis for phosphorylated p38 (p-p38) and p38 using proteins extracted from LPS-treated RAW264 cells (Figure 7B and Appendix A). Using densitometric analysis, we determined the relative levels of p-p38 and p38 (i.e., p-p38/p38) normalized against Gapdh levels (Figure 7C). The p-p38/p38 ratio was enhanced 1.2-fold when RAW264 cells were cultured in the presence of LPS (LPS only) compared with that in RAW264 cells cultured without LPS (i.e., LPS(−). In contrast, the p-p38/p38 ratio was suppressed in a concentration-dependent manner in RAW264 cells cultured with LPS and MDZ. Similarly, in another control, in cells cultured with LPS and SB203580, a p38MAPK inhibitor, the p-p38/p38 ratio decreased in a concentration-dependent manner. In particular, the p-p38/p38 ratio in cells cultured with LPS and 40 µM MDZ or with LPS and 10 µM SB203580 was significantly reduced relative to that in cells cultured with LPS alone (1.73-fold for LPS and 40 µM MDZ, and 1.75-fold for LPS and 10 µM SB203580).

## 3. Discussion

MDZ is one of the drugs listed on the WHO Model List of Essential Medicines (“WHO Model Lists of Essential Medicines—23rd list, 2023”, retrieved on 13 October 2023 from https://www.who.int/publications/i/item/WHO-MHP-HPS-EML-2023.02). To date, MDZ has been used as an adjunct to intravenous sedation in a wide range of diagnostic and therapeutic procedures, and is widely accepted by patients and clinicians. Research on drug repositioning of MDZ with guaranteed safety has been conducted considering its potential as a therapeutic agent for promoting bone formation [17]. In this study, we aimed to gain insights into the effects of MDZ on bone resorption and examined the potential of MDZ as a therapeutic agent for diseases associated with inflammatory bone resorption.

Osteoclast differentiation studies using RAW264 cells have shown that mature osteoclasts in TRAP-positive cells form within 5 days [20,21,22]. In agreement with this information, we found a concentration-dependent inhibition of TRAP activity and resorption pit formation to be associated with MDZ. Moreover, we found that MDZ suppressed the expression of osteoclast differentiation marker genes in a concentration-dependent manner. Taken together, our results indicate that MDZ may effectively inhibit osteoclast differentiation and reduce bone resorption capacity.

According to the National Library of Medicine at the National Institutes of Health, for perioperative use, the induction dose of MDZ for intravenous administration ranges from 0.15 to 0.40 mg/kg (“National Library of Medicine at National Institutes of Health, Midazolam, retrieved on 16 October 2023, from https://www.ncbi.nlm.nih.gov/books/NBK537321). In animal studies, we observed that MDZ administered at 2 mg/kg to a 40 g LPS-induced calvarial model mouse suppressed inflammatory bone resorption. Based on human-equivalent dose (HED) calculation (“Guidance for Industry Estimating the Maximum Safe Starting Dose in Initial Clinical Trials for Therapeutics in Adult Healthy Volunteers, Table V, “STEP 2: HUMAN EQUIVALENT DOSE CALCULATION, B. Basis for Using mg/kg Conversions, retrieved on 16 October 2023 from https://www.fda.gov/media/72309/download), the HED calculated using these values is 0.168–0.182 mg/kg for males and 0.183–0.192 mg/kg for females; both are within the induction dose range for MDZ. Thus, the safety concerns associated with MDZ use are alleviated.

LPS is a well-known cause of pathological bone resorption in periodontal and other diseases. Animal studies on LPS-induced osteoclastogenesis and bone resorption have revealed the effects of LPS on cranial bone resorption using micro-CT [23,24,25]. In this study, LPS was injected subcutaneously into the central calvaria of mice to establish a bone resorption model. Our results showed that MDZ inhibited LPS-induced cranial bone resorption in vivo, which highlights the potential for its application in diseases characterized by inflammatory bone destruction. Thus, both in vitro and in vivo results led us to investigate the mechanism involved in the inhibition of osteoclast differentiation by MDZ.

We focused on the involvement of ROS in mediating the MDZ action. LPS induces ROS generation in RAW264 cells [26]. ROS are generally classified into four types: superoxide radicals, hydroxyl radicals, hydrogen peroxide, and singlet oxygen. Superoxides, which are formed by a one-electron reduction of oxygen, are generated by ROS-producing enzyme systems, for example, NADPH oxidase (NOX), in phagocytic cells, such as neutrophils and macrophages [27]. NOX isoforms localize to the plasma and internal membranes, such as the endoplasmic reticulum [28] and the nuclear [29] and mitochondrial membranes [30]. When LPS is sensed by a Toll-like receptor 4 (TLR4), NOX4 in the mitochondrial inner membrane is activated, and ROS are produced [31,32]. During osteoclastogenesis, ROS are produced by RANKL through RANK-tumor necrosis factor (TNF)-receptor-associated factor 6 (TRAF6)-Rac1 signaling [33,34]. MDZ generally binds to the benzodiazepine (BDZ)-binding site on gamma-aminobutyric acid (GABA) A receptors, thereby increasing the action of GABA and permeability of chloride ions. Unlike the central BDZ receptor, peripheral BDZ receptors (PBRs) are present in peripheral tissues [35]. PBR is a mitochondrial protein, normally located in the mitochondrial outer membrane, and MDZ binding inhibits its production [36]. We demonstrated that ROS production was suppressed to control levels in the presence of MDZ in both RANKL- and LPS-treated cultures. This suggests that MDZ negatively regulates ROS production via mitochondrial PBR.

The p38 MAPK is one of the signaling pathways that play an important role in osteoclast differentiation [22,37]. When LPS is sensed by TLR4, ROS are produced in the mitochondria and activate apoptosis signal-regulating kinase 1 (ASK1) bound toTRAF6, a target of ROS. Following the activation of ASK1, downstream p38 is also activated [38,39]. In the present study, we found that MDZ inhibits p38 MAPK signaling at the protein level. Taken together with the previously mentioned results of suppression of ROS production by MDZ, the potential mechanism behind the suppression of osteoclast differentiation by MDZ in the present study, as shown in Figure 8, involves suppression of ROS production in mitochondria by MDZ via PBR and negative regulation of the TRAF6-p38 pathway induced by RANKL or LPS.

Thus, the present study shows that MDZ may inhibit bone resorption. However, the findings in RAW264, which is a macrophage cell line, might not reflect the actual bone resorption process. Therefore, it may be difficult to directly relate the results of in vitro experiments to the actual physiological conditions in vivo. Furthermore, it is not clear how MDZ affects other cells and tissues, and whether there are any side effects associated with its use. To overcome these limitations, investigation of the effects on other cells and tissues is warranted.

## 4. Materials and Methods

### 4.1. Biochemical and Cell Biology Experiments

#### 4.1.1. Preparation of Mouse Macrophage-like Cell Line (RAW264 Cells)

The mouse macrophage cell line RAW264 (RCB0535) was obtained from the RIKEN BioResource Center (Tsukuba, Ibaraki, Japan) under the National Bio-Resource Project of the Ministry of Education, Culture, Sport, Science and Technology, Japan. Experiments were performed using cells at passage three after cloning. The cells were suspended in a fresh standard medium composed of 10% fetal bovine serum (FBS), 89% alpha-minimum essential medium Eagle alpha modification (Gibco/Thermo Fisher Scientific, Waltham, MA, USA), and 1% antibiotics (5000 U/mL penicillin, 5 mg/mL streptomycin, Gibco/Thermo Fisher Scientific, Waltham, MA, USA). They were then seeded in 96-well plates at a density of 5.0 × 10^3^ cells/well for TRAP assay or in 12-well plates at a density of 5.0 × 10^4^ cells/well for TRAP staining, and then cultured in standard medium under a humidified 5% CO_2_ atmosphere for 24 h at 37 °C. The medium was replaced with a growth medium, prepared by supplementing 0, 5, 10, 20, or 40 μM MDZ (Merck, Darmstadt, Germany) and 300 ng/mL glutathione S-transferase-receptor activator of NF-kappa B ligand (GST-RANKL) in standard medium, and the cells were cultured for an additional 3 days.

#### 4.1.2. TRAP Assay

The cells were rinsed with phosphate-buffered saline, and the TRAP activity was evaluated after incubating the cells with 0.1 mL/well of TRAP buffer (0.1 M sodium acetate, 50 mM sodium tartrate, pH 5.0), containing 0.005% Triton X and 10 mM *p*-nitrophenylphosphate as a substrate, at 20 °C for 10 min. The reaction was quenched by adding half the volume of 0.2 M NaOH, and the absorbance at 405 nm was measured using a plate reader (iMark microplate reader, Bio-Rad, Hercules, CA, USA).

#### 4.1.3. TRAP Staining of RAW264 Cells

The cells were rinsed with PBS, fixed in 4% paraformaldehyde at room temperature for 5 min, rinsed with distilled water, and then incubated for 30 min at room temperature in TRAP buffer (pH 5.0) containing 0.1 mg/mL of naphthol AS-MX phosphate as a substrate and 0.6 mg/mL of fast red violet LB salt (TRAP stain solution). After removing the TRAP stain solution and rinsing the cells with PBS, the cell morphology was observed using an optical microscope (CKX53, OLYMPUS, Tokyo, Japan) equipped with a DP22 camera (OLYMPUS).

#### 4.1.4. Pit Formation Assay

Osteoclast-mediated bone resorptive activity was quantified using a pit assay performed in a CaP-coated plate [40]. Solutions of Na_2_HPO_4_ (0.12 M) and CaCl_2_ (0.2 M; in 50 mM Tris-HCl, pH 7.4) were sterilized with a syringe filter (0.2 μm pore size) and mixed in equal amounts. The cloudy mixture that was obtained was incubated for 4 h at room temperature. The precipitated CaP slurry was washed five times with sterile distilled water and its volume was adjusted with sterile water to 12 times the original volume. Aliquots of the CaP slurry (100 µL) were dispensed into each well of a 96-well culture plate. The culture plate was dried at 37 °C for two days. RAW264 cells were plated on CaP-coated plates at a density of 5.0 × 10^3^ cells/well and cultured in the standard medium. One day after plating, the culture medium was replaced with fresh medium containing 300 ng/mL GST-RANKL. After an additional 4 days of culture, the plate was washed with PBS and the cells were removed from the wells by treating with 1 M ammonium solution for 10 min. The plate was washed three times with distilled water and dried thoroughly. The image of the CaP-coated plate was captured using an EPSON GT-X980 scanner (Seiko Epson Corp., Suwa, Nagano, Japan), and the area of pits formed on the bottom of the wells by osteoclasts was measured using the ImageJ software (v.1.52a) (National Institutes of Health, Bethesda, MD, USA).

#### 4.1.5. Quantitative Polymerase Chain Reaction Analysis

Quantitative polymerase chain reaction was performed using the SYBR Green technique on a LightCycler 96 System (Roche Diagnostics GmbH, Mannheim, Germany). Specific primer sets were designed using a primer-designing tool, Primer-BLAST (National Institutes of Health, Bethesda, MD, USA). Glyceraldehyde-3-phosphate dehydrogenase (*Gapdh*) was used as a reference gene. The specific PCR primer sets and reaction conditions are listed in Appendix A. The expression levels of *Nfatc1*, *Trap*, and *Ctsk* genes were normalized to that of *Gapdh*.

### 4.2. Animal Experiments

#### 4.2.1. Establishment of Lipopolysaccharide-Induced Calvarial Model Mouse and TRAP Staining

Eighteen nine-week-old male Institute of Cancer Research (ICR) mice (CLEA Japan, Inc., Tokyo, Japan) (40 g body weight) were divided into three groups: Group 1: PBS alone (control) (*n* = 6); Group 2: LPS-only (*n* = 6); and Group 3: combination of LPS and MDZ (LPS+MDZ) (*n* = 6). Mice were anesthetized by isoflurane inhalation every 2 days and inoculated at the intersection of the sagittal and coronal sutures with PBS (2 mL/kg; for group 1), LPS (5 mg/kg; for group 2), or LPS and MDZ (5 and 2 mg/kg, respectively; for group 3). Randomization was not used to assign experimental units to the control and treatment groups. In addition, confounders were not controlled. All animals were weighed on days 0, 2, 4, 6, and 7 and euthanized under CO_2_ anesthesia on day 7. The calvariae were sampled, immediately washed with PBS, and fixed in 4% paraformaldehyde. For TRAP staining, the calvaria in 4% paraformaldehyde were rinsed with distilled water and incubated for 30 min at 20 °C in TRAP stain solution. They were then visualized under a stereomicroscope (SZX7, OLYMPUS).

#### 4.2.2. Micro-Computed Tomography (Micro-CT) and Quantitative Analysis of Suture

The calvarial specimens were scanned using a micro-computed tomography (micro-CT) system (InspeXio SMX-225CT; Shimadzu, Kyoto, Japan). The scanning conditions were as follows: tube voltage, 105 kV; tube current, 70 μA; resolution, 1024 × 1024 pixels; operating conditions, 1200 views; and total scanning time, 900 s. The micro-CT images were analyzed using a three-dimensional reconstruction software (TRI/3D-BON-FCS64; Ratoc, Tokyo, Japan), and the width and bone matrix volume of each suture were analyzed. To measure the width of the suture, 10 locations were selected for each suture, and the average width of each group was calculated using the ImageJ software. For measuring the bone matrix volume, blocks of lambdoid suture (1120 μm length × 4000 μm width × 365 μm height per block) and coronal sutures (1600 μm length × 1600 μm width × 231 μm height per block) were created for each individual, and the bone matrix volume of each block was measured. Both the width and bone matrix volume were measured for six individuals following the same procedure.

### 4.3. Measurement of Reactive Oxygen Species

RAW264 cells were plated in 96-well plates at a density of 4.0 × 10^5^ cells/well and cultured in the standard medium under a humidified 5% CO_2_ atmosphere for 24 h at 37 °C. The next day, reactive oxygen species (ROS) levels were measured using an ROS Detection Cell-Based Assay Kit (DHE) (Cayman Chemical, Ann Arbor, MI, USA). The culture medium was aspirated and 150 µL of Cell-Based Assay buffer was added; 130 µL of buffer was immediately aspirated and 130 µL of ROS Staining buffer supplemented with or without 10 µL of MDZ, 10 µL of RANKL, or 10 µL of LPS was added. In addition to the MDZ, RANKL, and LPS, 10 µL of N-acetyl cysteine or 10 µL of antimycin A was added in ROS Staining buffer as negative or positive control. The cells were incubated for 90 min at 37 °C, and the ROS Staining buffer was replaced with 100 µL of Cell-Based Assay buffer. ROS levels were measured with a fluorescence plate reader (Skanlt RE for Varioskan Flash 2.4, Thermo Fisher Scientific) using excitation and emission wavelengths of 480 and 570 nm, respectively.

### 4.4. Western Blot Analysis

RAW264 cells were seeded in a six-well plate at a density of 3 × 10^5^ cells/well and cultured in the standard medium under a humidified 5% CO_2_ atmosphere for 24 h at 37 °C. The medium was changed to the standard medium supplemented with or without LPS (100 ng/mL), MDZ (20 or 40 µM), or SB203580 (1 or 10 µM) (AdipoGen, San Diego, CA, USA), and the cells were cultured for 15 min. Proteins were extracted from the cultured cells using RIPA buffer to prepare samples for Western blotting. Sodium dodecyl sulphate-polyacrylamide gel electrophoresis was performed on 5–20% e-PAGEL minigels (ATTO Corporation, Tokyo, Japan). The proteins were transferred from the gel onto an Invitrolon polyvinylidene fluoride membrane (Life Technologies/Invitrogen/Thermo Fisher Scientific, Carlsbad, CA, USA). The blots were incubated with p38 mitogen-activated protein kinase (p38) (#9212, Cell Signaling, Danvers, MA, USA) and phosphorylated p38 (p-p38) (#9211, Cell Signaling) polyclonal antibodies (1:1000, 20 h) or with glyceraldehyde-3-phosphate dehydrogenase (Gapdh) antibody (STJ140038, St. John’s Laboratory, London, UK) (1:1000, 20 h) at 4 °C. Goat anti-rabbit IgG (H+L) HRP-conjugated antibody (#172-1019, Bio-Rad Laboratories) (1:2000, 1 h at 23 °C) for p38 and p-p38, and rabbit anti-goat IgG (H+L) HRP-conjugated antibody (#172-1034, Bio-Rad Laboratories) (1:2000, 1 h at 23 °C) for Gapdh were used as secondary antibodies. Bands were detected using a chemiluminescence substrate, ECL Prime (GE Healthcare, Chicago, IL, USA), and the intensity of the detected p38 and p-p38 bands was quantified using the ImageJ software, with Gapdh intensity used as a standard. The relative levels of p-p38 and p38 (i.e., p-p38/p38) were calculated from the quantitative values obtained for the p38 and p-p38 bands. Western blot analysis was performed via three independent experiments.

### 4.5. Statistical Analysis

Statistical significance was determined using the nonparametric Steel’s test for the measurement of body weight, TRAP assay, and pit formation assay, and the Steel–Dwass test for qPCR analysis, measurement of suture width, and measurement of bone matrix volume around the suture. Data in parentheses indicate the 25–75% point of the interquartile range, and the solid line within the parentheses indicates the median. Data for body weight and quantitative data obtained in the Western blot analysis are presented as the means ± standard error of the mean (SEM). For all data, a *p*-value < 0.05 was considered to indicate a statistically significant difference.

## 5. Conclusions

In the present study, the inhibitory effect of MDZ on bone resorption was demonstrated and its mechanism was elucidated. Our findings suggest that MDZ is a potential therapeutic agent for inflammatory bone destruction diseases, such as periodontal disease. However, further studies using animal models of disease, validation of combined effects with existing drugs, and clinical trials are required to clarify the potential contribution of MDZ as a novel therapy. In addition, the effects of MDZ on bone metabolism and its safety should be evaluated in greater detail.

## Figures and Tables

**Figure 1 ijms-25-07651-f001:**
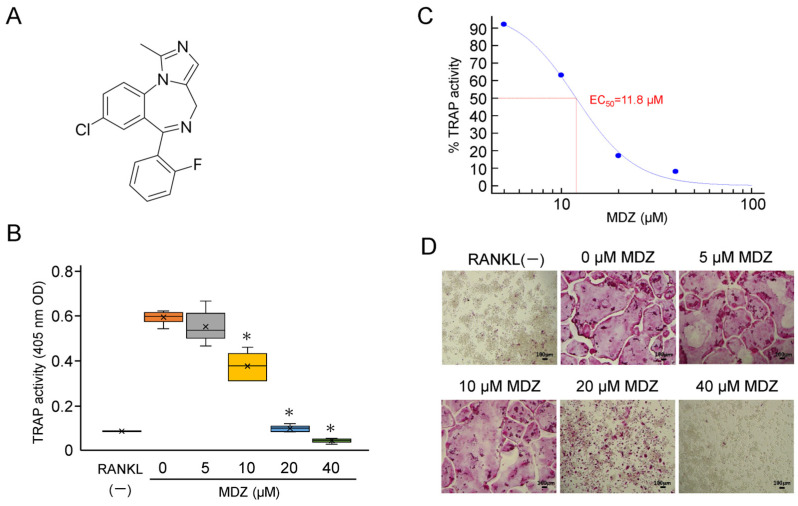
Effect of MDZ on osteoclast differentiation of RAW264 cells. (**A**) Structure of MDZ. (**B**) TRAP activity in the absence of RANKL and MDZ (RANKL(−)) and in the presence of 300 ng/mL RANKL together with MDZ (0, 5, 10, 20, and 40 μM). For each group, the experiment was carried out at the same time and repeated six times (*n* = 6). The TRAP activity was determined in triplicate for each group of samples. The asterisks (*) indicate significant differences compared with the activity at 0 µM MDZ (* *p* < 0.05, nonparametric Steel’s test). (**C**) Sigmoid curve created based on dose–response data of TRAP activity used for EC_50_ calculation. The activity values in the 5, 10, 20, and 40 μM MDZ treatments are shown relative to that in the absence of MDZ (0 μM), which was set to 100%. The calculated EC_50_ value is shown in red. (**D**) Representative optical microscopic images of TRAP-stained RAW264 cells on day 3 of culture after the addition of MDZ. Observations were performed using a 10× objective lens. The formation of multinucleated osteoclasts was suppressed in the presence of 20 and 40 μM MDZ (Scale bar = 100 μm). MDZ: midazolam; RANKL: receptor activator of nuclear factor kappa B ligand; TRAP: tartrate-resistant acid phosphatase.

**Figure 2 ijms-25-07651-f002:**
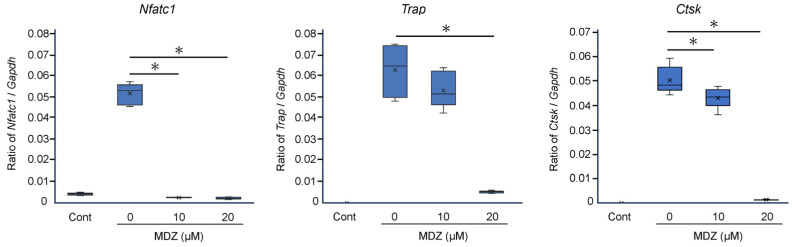
Effect of MDZ on the expression of osteoclast differentiation marker genes in RAW264 cells. RAW264 cells were cultured in the absence of RANKL and MDZ (Cont) or in the presence of 300 ng/mL RANKL together with MDZ (0, 10, and 20 μM). *Nfatc1*: nuclear factor of activated T cells 1; *Trap*: tartrate-resistant acid phosphatase; and *Ctsk*: cathepsin K. The level of each mRNA was normalized to that of the reference gene glyceraldehyde-3-phosphate dehydrogenase (*Gapdh*), and the relative quantification data for *Nfatc1*, *Trap*, and *Ctsk* mRNA levels in RAW264 cells were generated based on a mathematical model for relative quantification in the qPCR system. The asterisk (*) indicates a significant difference in values between each concentration of MDZ (* *p* < 0.05, Steel–Dwass test). MDZ: midazolam.

**Figure 3 ijms-25-07651-f003:**
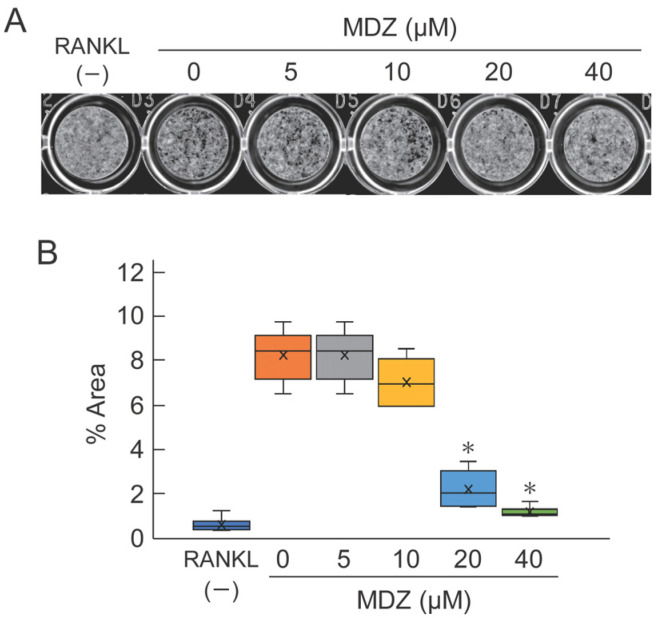
Effect of MDZ on osteoclast-mediated bone resorption as determined using the pit formation assay. (**A**) Representative images of pits. Images of the bottom of CaP-coated plate were captured with a scanner and the resorption pits (black regions) were observed. The formation of resorption pits was suppressed in the presence of 20 and 40 μM MDZ. (**B**) Percentage area of resorption pits. The area of the black regions was summed using the ImageJ software. For each group, the experiment was carried out at the same time and repeated six times (*n* = 6). The asterisks (*) indicate significant differences compared with the value obtained at 0 µM MDZ (* *p* < 0.05, nonparametric Steel’s test). MDZ: midazolam.

**Figure 4 ijms-25-07651-f004:**
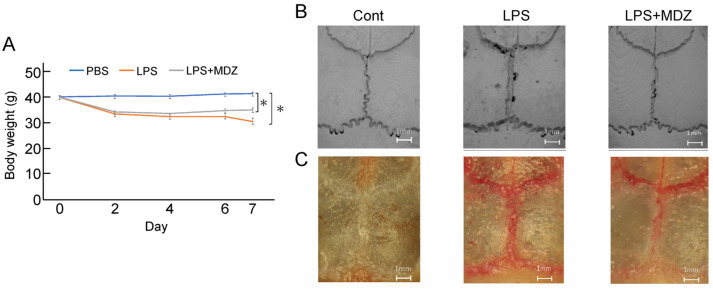
Morphological observations of the effect of MDZ on LPS-induced calvarial mouse model. (**A**) Changes in the body weight of mice during the treatment period. Values are the means ± standard error of the mean (SEM) for six mice (* *p* < 0.05, Steel–Dwass test). (**B**) Representative micro-CT 3D reconstruction images of mouse calvaria (Scale bar = 1 mm). (**C**) Representative stereomicroscopic images of TRAP staining showing osteoclastogenesis in each sample. Cont: PBS-alone dosage group (i.e., control); LPS: LPS-only dosage group; LPS+MDZ: combination of LPS and MDZ dosage group (Scale bar = 1 mm). LPS: lipopolysaccharide; MDZ: midazolam; micro-CT: micro-computed tomography; TRAP: tartrate-resistant acid phosphatase.

**Figure 5 ijms-25-07651-f005:**
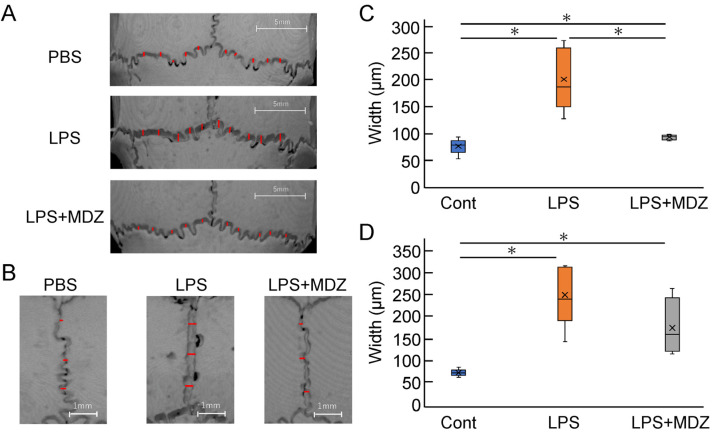
Validation of suture width on bone destruction in an LPS-induced calvarial mouse model. (**A**,**B**) Locations of lambdoidal and sagittal sutures, respectively, with red lines indicating measurement sites. Scale bars = 5 mm for (**A**) and 1 mm for (**B**). (**C**,**D**) Average width of lambdoidal and sagittal sutures for each group. Measurements were obtained at ten (lambdoidal) and three (sagittal) sites per suture using the ImageJ software (v.1.52a). Six mice per group (*n* = 6). Significant differences are indicated by asterisks (* *p* < 0.05, Steel–Dwass test). Cont: PBS-alone (i.e., control); LPS: LPS-only; LPS+MDZ: LPS and MDZ combination. LPS: lipopolysaccharide; MDZ: midazolam; PBS, phosphate-buffered saline.

**Figure 6 ijms-25-07651-f006:**
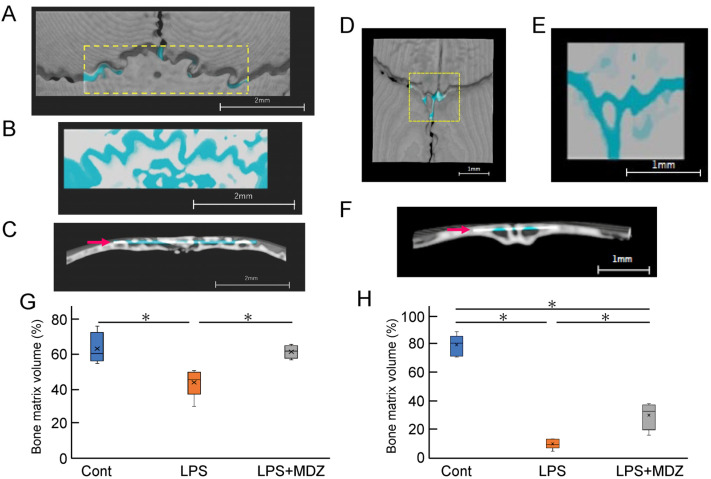
Validation of bone matrix volume on bone destruction in LPS-induced calvarial mice. (**A**,**D**) Representative images showing the location of lambdoidal (**A**) and coronal (**D**) sutures for measurement of bone matrix volume. For each mouse, a block of lambdoidal (1120 µm long × 4000 µm wide) and coronal (1600 µm long × 1600 µm wide) suture in the area indicated by the yellow dotted box was created using a 3D reconstruction software (TRI/3D-BON-FCS64, RATOC, Tokyo, Japan). Scale bars = 2 mm for (**A**) and = 1 mm for (**D**). (**B**,**E**) Representative images of the upper part block of lambdoidal (**B**) and coronal (**E**) sutures. Bone matrix area is shown in white or gray and seam or bone destruction areas are shown in light blue. Scale bars = 2 mm for (**B**) and = 1 mm for (**E**). (**C**,**F**) Representative images of the lateral block of lambdoidal (**C**) and coronal (**F**) sutures. The thickness of each block, indicated by the red arrow, was 365 µm for lambdoidal suture and 231 µm for coronal suture. Scale bars = 2 mm for (**C**) and = 1 mm for (**F**). (**G**,**H**) Average bone matrix volume of lambdoidal (**G**) and coronal (**H**) sutures in each group. The average bone matrix volume for each group was calculated using the 3D reconstruction software. The measurement for each group was performed independently for six mice (*n* = 6). The asterisks (*) indicate significant differences between each group (* *p* < 0.05, Steel–Dwass test). Cont: PBS-alone dosage group (i.e., control); LPS: LPS-only dosage group; LPS+MDZ: combination of LPS and MDZ dosage group. LPS: lipopolysaccharide; MDZ: midazolam.

**Figure 7 ijms-25-07651-f007:**
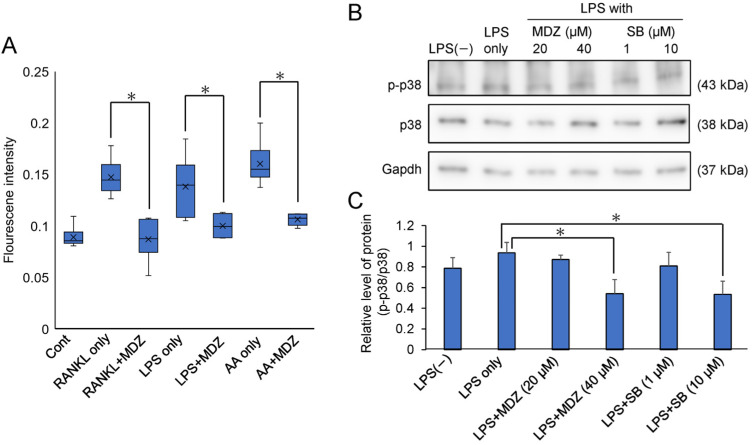
Effects of MDZ on reactive oxygen species (ROS) production and phosphorylated p38 levels in RAW264 cells. (**A**) Intracellular ROS levels in RAW264 cells. Fluorescence signals were measured using an ROS Detection Cell-Based Assay Kit (DHE) (Ex/Em = 480/570 nm). RANKL: receptor activator of nuclear factor kappa B ligand; MDZ: midazolam; LPS: lipopolysaccharide; AA: antimycin A. Cont: control (i.e., RAW264 cells cultured alone without any substances); RANKL only, LPS only, and AA only: RAW264 cells cultured with RANKL alone, LPS alone, and AA alone, respectively. ROS analysis was performed via six independent experiments (*n* = 6). The asterisks (*) indicate significant differences between each group (* *p* < 0.05, nonparametric Steel’s test). (**B**) Western blotting images showing the levels of p-p38 (**top**), p38 (**middle**), and Gapdh (**bottom**) in RAW264 cells treated with LPS. p38: p38 mitogen-activated protein kinase; p-p38: phosphorylated p-38; Gapdh: glyceraldehyde-3-phosphate dehydrogenase. LPS: lipopolysaccharide; MDZ: midazolam; SB: SB203580. LPS(−): RAW264 cells cultured without LPS; LPS only: RAW264 cells cultured with LPS alone. (**C**) Analysis of the relative level of p-p38 and p38 normalized against Gapdh levels in LPS-treated RAW264 cells using densitometry of Western blots, employing the ImageJ software (v.1.52a). Western blot analysis was performed via three independent experiments (*n* = 3). The asterisks (*) indicate significant differences between the LPS-only and each group (* *p* < 0.05, Student’s *t*-test). LPS: lipopolysaccharide; MDZ: midazolam; SB: SB203580. LPS(−): RAW264 cells cultured without LPS; LPS only: RAW264 cells cultured with LPS alone.

**Figure 8 ijms-25-07651-f008:**
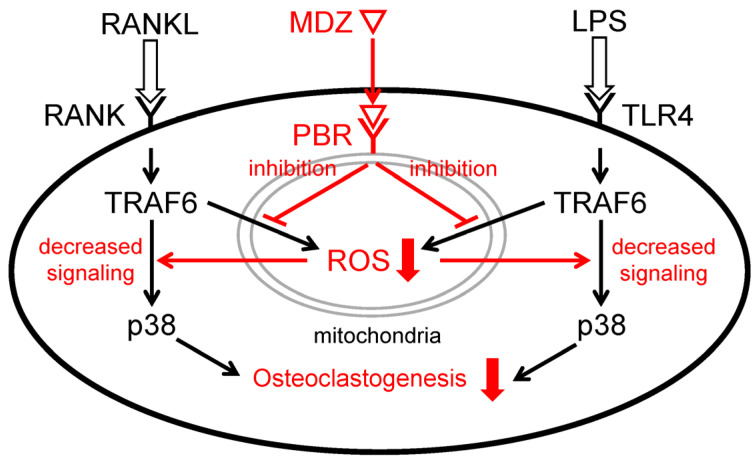
Schematic diagram illustrating the mechanism of inhibition of RANKL- or LPS-induced osteoclast differentiation by MDZ. MDZ suppresses the production of ROS in mitochondria via PBR and negatively regulates the TRAF6-p38 pathway induced by RANKL or LPS. As a result, differentiation into osteoclasts is inhibited. RANKL: receptor activator of nuclear factor kappa B ligand; RANK: receptor activator of nuclear factor kappa B ligand; TRAF6: tumor necrosis factor receptor-associated factor 6; p38: p38 mitogen-activated protein kinase; MDZ: midazolam; PBR: peripheral benzodiazepine receptor; ROS: reactive oxygen species; LPS: lipopolysaccharide; TLR4: Toll-like receptor 4.

## Data Availability

All data presented in this article are available in the manuscript.

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
