# Peer review of "Potential for Drug Repositioning of Midazolam as an Inhibitor of Inflammatory Bone Resorption"

_ijms, 2024, doi:10.3390/ijms25147651_

Round 1

Reviewer 1 Report

Comments and Suggestions for Authors

This manuscript aims to examine the effects of MDZ on bone resorption at the biochemical, cellular, molecular, and morphological levels using cell-based in vitro and animal experiments.

Hallmarks of this manuscript are: rich contents, and sufficient information. Furthermore, this manuscript displays a  densely organized structure. Anyway, there are some sections of this manuscript that should be improved. For these reasons, the manuscript requires major changes.

Please find below an enumerated list of comments on my review of the manuscript:

MINOR POINTS:

There is lack of a list of the abbreviations, mentioned in this manuscript. Please, if possible, provide it.

MAJOR POINTS:

INTRODUCTION:

LINE 39: These processes are regulated by the involvement of different factors, from immunity cells, to growth factors, and proteins, which contribute to the balance and progression of the cellular and molecular dynamics, underlying bone resorption and synthesis (see, for reference: https://doi.org/10.23812/j.biol.regul.homeost.agents.20233705.232). This is the major point of this manuscript: according to recent evidence on this topic, the authors should mention the multifactorial contribute of different cellular and molecular elements in the onset and progression of processes, associated to bone resorption and formation. When added, a brief overview of this issue will improve the impact and quality of this manuscript.

LINE 45: The treatment of periodontitis is multi-staged and first steps of non-surgical periodontal therapy (NSPT) include supragingival plaque control, followed by subgingival scaling and root planing (SRP), aimed at eliminating biofilms, endotoxins, and calculus (see, for reference: https://doi.org/10.1007%2Fs00784-024-05674-7).

LINE 66: Here, we examined the effects of MDZ on bone resorption at the biochemical, cellular, molecular, and morphological levels using cell-based in vitro and animal experiments.

RESULTS:

Figure 1: The authors should add the magnification of the images of TRAP-stained in the figure label, also mentioning the type of microscopical techniques applied.

As regards the section of methods, there is a specific and detailed explanation for the methods used in this study: this is particularly significant, since the manuscript relies on a multitude of methodological and statistical analysis, to derive its conclusions. The methodology applied is overall correct, the results are reliable and adequately discussed.

Besides, the methodology design was appropriately implemented within the study. However, many of the topics are very concisely covered. However, major concerns of this manuscript are with the introduction and results sections: for these reasons, I have major comments for these sections, for improvement before acceptance for publication. The article is accurate and provides relevant information on the topic and I have some major points to make, that may help to improve the quality of the current manuscript and maximize its scientific impact. I would accept this manuscript if the comments are addressed properly.

Author Response

Response to Reviewer 1 Comments

Thank you very much for taking the time to review this manuscript. Please find the detailed responses below and the corresponding revisions/corrections highlighted in the re-submitted files.

MINOR POINTS:

Comments 1: There is lack of a list of the abbreviations, mentioned in this manuscript. Please, if possible, provide it.

Response 1: Thank you for highlighting this issue. In accordance with your suggestion, a list of abbreviations has been included in the revised manuscript (line 514).

MAJOR POINTS:

INTRODUCTION

Comments 2: LINE 39: These processes are regulated by the involvement of different factors, from immunity cells, to growth factors, and proteins, which contribute to the balance and progression of the cellular and molecular dynamics, underlying bone resorption and synthesis (see, for reference: https://doi.org/10.23812/j.biol.regul.homeost.agents.20233705.232). This is the major point of this manuscript: according to recent evidence on this topic, the authors should mention the multifactorial contribute of different cellular and molecular elements in the onset and progression of processes, associated to bone resorption and formation. When added, a brief overview of this issue will improve the impact and quality of this manuscript.

Response 2: Thank you for your very valuable comments. Given that our study has identified the signaling associated with MDZ inhibition of bone resorption, we believe the text suggested by the reviewer is crucial. Therefore, we have added the following text to lines 40–44 and cited recommended references. "In the onset and progression of these bone resorption- and bone formation-related processes, various cellular and molecular elements, ranging from immune cells to growth factors and proteins, play multifactorial riles and contribute to the balance and progression of cellular and molecular dynamics underlying bone resorption and bone formation [4]."

Comments 3: LINE 45: The treatment of periodontitis is multi-staged and first steps of non-surgical periodontal therapy (NSPT) include supragingival plaque control, followed by subgingival scaling and root planing (SRP), aimed at eliminating biofilms, endotoxins, and calculus (see, for reference: https://doi.org/10.1007%2Fs00784-024-05674-7).

Response 3: Thank you for your valuable comments. As you pointed out, the explanation for the NSPT was missing. The relevant information has been added to the revised manuscript (lines 50-52). "Treatment of periodontitis is a multi-step process, and the first step in nonsurgical periodontal therapy includes plaque control above the gingival margin, followed by subgingival scaling and root planing to remove biofilm, endotoxins, and calculus [7].

Comments 4: LINE 66: Here, we examined the effects of MDZ on bone resorption at the biochemical, cellular, molecular, and morphological levels using cell-based in vitro and animal experiments.

Response 4: Thank you for pointing this out. We have removed sentences from lines 77 to 79.

RESULTS

Comments 5: Figure 1: The authors should add the magnification of the images of TRAP-stained in the figure label, also mentioning the type of microscopical techniques applied.

Response 5: Thank you for highlighting this issue. Regarding the microscopic technique, we have added "optical microscopic" to the legend (line 104) in Figure 1(D). In addition, we have added the following explanation to lines 105-106: “Observations were performed using a 10× objective lens.”

As regards the section of methods, there is a specific and detailed explanation for the methods used in this study: this is particularly significant, since the manuscript relies on a multitude of methodological and statistical analysis, to derive its conclusions. The methodology applied is overall correct, the results are reliable and adequately discussed.

Besides, the methodology design was appropriately implemented within the study. However, many of the topics are very concisely covered. However, major concerns of this manuscript are with the introduction and results sections: for these reasons, I have major comments for these sections, for improvement before acceptance for publication. The article is accurate and provides relevant information on the topic and I have some major points to make, that may help to improve the quality of the current manuscript and maximize its scientific impact. I would accept this manuscript if the comments are addressed properly.

Again, we would like to thank the reviewer for the supportive comments and helpful suggestions.

Reviewer 2 Report

Comments and Suggestions for Authors

Dear authors,

The paper is interesting, but I have some flaws.

Introduction

Line 25-26: Thus, we propose that MDZ could potentially be used for treating inflammatory bone resorption, for example, in periodontal disease.

What is the hypothesis of the study?

2. Results

Fig. 5. Ten locations on the lambdoidal suture and three locations on the sagittal suture was measured for each group.

How did you select the locations for morphological observation?

Why did the author not performed slides stained with hematoxylin-eosin?

Why did the author not analyze tissue anatomical pathologic changes such as inflammation?

The legends of the figures are very extensive.

3- Discussion

Line 276-281:

This text is unnecessary.

4. Materials and Methods

4.2 Animal experiments

The authors used a mouse model.

ARRIVE guidelines is missing in the experiment.

How many animals are used? How did you determined the number of the animals used in each group and the time of the experimental period?

Line 413-414: The authors inoculated on the head with PBS, LPS or LPS and MDZ

Which region of the skull was inoculated?

4.2.2 Micro-computed tomography

How many mice did you used for micro-CT tomography?

Author Response

Response to Reviewer 2 Comments

Thank you very much for taking the time to review this manuscript. Please find the detailed responses below and the corresponding revisions/corrections highlighted in the re-submitted files.

1.       Introduction

Comments 1: Line 25-26: Thus, we propose that MDZ could potentially be used for treating inflammatory bone resorption, for example, in periodontal disease.

What is the hypothesis of the study?

Response 1: Thank you for your question. As described in “Introduction” lines 68-71, in our previous study on MDZ, we reported that MDZ exerts an osteogenic effect. Therefore, given that bone remodeling occurs in parallel with bone formation and resorption, we hypothesized that MDZ may also exert some effects on bone resorption. In addition, "MDZ has been shown to suppress the production of interleukin-6 in human peripheral blood mononuclear cells and may suppress immune and inflammatory responses [18]." This report further encouraged our hypothesis and led us to clarify the action of MDZ on inflammatory bone resorption, the subject of the current study. Therefore, we have added a sentence to the “introduction” in lines 72–74, citing the literature explaining the above.

2.  Results

Fig. 5. Ten locations on the lambdoidal suture and three locations on the sagittal suture was measured for each group.

Comments 2: How did you select the locations for morphological observation?

Response 2: Thank you for your question. Owing to individual differences in experimental animals, normalization was performed by measuring multiple locations at random.

Comments 3: Why did the author not performed slides stained with hematoxylin-eosin?

Response 3: Thank you for your question. Given that this reply is also relevant to the following question, we have jointly replied to both queries.

Comments 4: Why did the author not analyze tissue anatomical pathologic changes such as inflammation?

Response 4: Thank you for your question. Sections need to be prepared for histological comparisons. We attempted to prepare sections and make observations; however, it is necessary to make observations over time to analyze changes in inflammation. In addition, although all experimental mouse groups comprised the same species, the shape of the skull varies among individuals. Furthermore, quantitatively analyzing the process of demineralization associated with section preparation and bone loss due to bone resorption (by LPS) can be challenging. Therefore, in this experiment, we used µCT, which is non-destructive and can perform partial analysis (as shown in Figure 6) to account for individual differences. Therefore, histological comparisons of hematoxylin-eosin-stained sections were not performed in this study.

Comments 5: The legends of the figures are very extensive.

Response 5: Thank you for highlighting this. To improve the conciseness of the legend, we incorporated a few changes and revised the description as follows (lines 170-177). “Figure 5. Validation of suture width on bone destruction in an LPS-induced calvarial mouse model. (A) and (B) Locations of lambdoidal and sagittal sutures, respectively, with red lines indicating measurement sites. Scale bars = 5 mm for (A) and 1 mm for (B). (C) and (D) Average width of lambdoidal and sagittal sutures for each group. Measurements were obtained at ten (lambdoidal) and three (sagittal) sites per suture using the ImageJ software (v.1.52a). Six mice per group (n = 6). Significant differences are indicated by asterisks (*p < 0.05, Steel–Dwass test). Cont: PBS-alone (i.e., control); LPS: LPS-only; LPS+MDZ: LPS and MDZ combination. LPS: lipopolysaccharide; MDZ: midazolam; PBS, phosphate-buffered saline.”

3.  Discussion

Comments 6: Line 276-281: This text is unnecessary.

Response 6: Thank you for highlighting this issue. In accordance with the above suggestion,  the sentences from lines 280 to 286 have been deleted.

  1. Materials and Methods

4.2 Animal experiments

The authors used a mouse model.

Comments 7: ARRIVE guidelines is missing in the experiment.

Response 7: Thank you for highlighting this issue. We have checked ARRIVE guidelines and uploaded it as “Non-published Material”. In addition, we added certain statements related to this guideline to “4.2.1” in lines 411 and 417-418 of Methods and “2.4.1” in lines 151-152 of Results. Also, we added the sentence to read, “In addition, all animal experiments were performed in accordance with ARRIVE guidelines.” In lines 505-506 of Institutional Review Board Statement.

Comments 8: How many animals are used? How did you determined the number of the animals used in each group and the time of the experimental period?

Response 8: Thank you for your question. As we described in “4.2.1” (lines 411-414), we used 18 mice (6 for control, 6 for LPS-only, and 6 for LPS and MDZ combination). Moreover, the experiment was performed for one week (the results were the same for one, two, or three weeks), with no difference observed in results when the duration was extended beyond one week.

Comments 9: Line 413-414: The authors inoculated on the head with PBS, LPS or LPS and MDZ

Which region of the skull was inoculated?

Response 9: Thank you very much for your question. We inoculated LPS at the intersection of the sagittal and coronal sutures. We have replaced the relevant statement (line 415).

4.2.2 Micro-computed tomography

Comments 10: How many mice did you used for micro-CT tomography?

Response 10: Thank you for your question. As previously stated in response 8, we used 18 mice (6 for control, 6 for LPS-only and 6 for LPS and MDZ combination).

Again, we would like to thank the reviewer for the supportive comments and helpful suggestions.

Round 2

Reviewer 1 Report

Comments and Suggestions for Authors

Congratulations to the authors, which have significantly improved the impact and quality of this manuscript. 

Reviewer 2 Report

Comments and Suggestions for Authors

Dear authors,

Thanks to provide a revised version of the paper with the suggested changes.

Best regards